# Transformers as Unrolled Inference in Probabilistic Laplacian Eigenmaps

**Aditya Ravuri**
University of Cambridge
aditya.ravuri@cl.cam.ac.uk

**Neil D. Lawrence**
University of Cambridge

## Abstract

We propose a probabilistic interpretation of transformers as unrolled inference steps, assuming an approximate probabilistic Laplacian Eigenmaps model from the ProbDR framework. Our derivation shows that at initialization, transformers perform "linear" dimensionality reduction. We also show that within the transformer block, a graph Laplacian term arises from our arguments rather than an attention matrix (which we interpret as an adjacency matrix). We demonstrate that simply subtracting the identity from the attention matrix (and thereby taking a graph diffusion step) improves validation performance on a language model and a simple vision transformer.

## 1  Introduction

Transformers, introduced in Vaswani et al. (2017), have been an incredibly successful architecture for deep learning, leading to vastly scaled models used for language as part of Large Language Models (LLMs), such as BERT (Devlin et al., 2019), vision transformers (ViTs) (Dosovitskiy et al., 2021), and foundation models for speech (e.g., wav2vec) (Baevski et al., 2020), as well as in many other application areas.

The mathematical basis for their success is an active area of interest. Good generalization cannot be achieved without some assumptions about the underlying data distribution. In this paper, we show that transformers can be seen to perform probabilistic *dimensionality reduction*. Dimensionality reduction enables generalization by imposing a lower dimensional manifold structure on the high dimensional data. Our mathematical approach is heavily inspired by the white-box transformer of Yu et al. (2023), who show that transformers can be viewed as unrolled inference assuming a mixture of Gaussians on latent representations. We provide an alternate interpretation, arguing that each block of a single-head transformer, at initialization, performs gradient descent on a variational lower bound of the probabilistic Laplacian Eigenmaps model of Ravuri et al. (2023). As part of a visual experiment, we show that MNIST flattened images cluster tightly by class when input to a transformer.

We use our interpretation of the transformer algorithm to guide us in improving its generalization performance by showing that a modification—simply subtracting an identity matrix from the attention matrix (in other words, performing graph diffusion or Laplacian smoothing in the attention step)—follows from our interpretation. We show that this architectural change can be more performant on a language model and vision transformer fit on the tiny Shakespeare and OpenWebText datasets (Karpathy, 2015; Gokaslan et al., 2019), and the downsampled Imagenet datasets (Russakovsky et al., 2015; Chrabaszcz et al., 2017) respectively. This work is a proof of concept on how the insights of Ravuri et al. (2023) can be used to improve engineering tools.

**Related Work**

In our work, we interpret the attention matrix as an adjacency matrix of a nearest-neighbor graph and show that unrolled optimization in a dimensionality reduction model leads to the transformer architecture.

The interpretation of attention matrices as matrices of data-point similarity or relevance has a long history; Vaswani et al. (2017) and many works since, for instance, Weng (2018); Chefer et al. (2021), have visualized attention matrices corresponding to text inputs, image patches, etc., for the purposes of interpretability. Recent work has interpreted the attention matrix as an adjacency matrix and shown that graph convolutions improve the performance of the architecture (Choi et al., 2024). In our work, we show that the graph diffusion steps also increase the performance of the architecture. In the realm of graph convolutional networks, Kipf & Welling (2017) motivate their architecture from a spectral graph convolutional perspective, and using a slightly different derivation of their updates, we find that an update also involves a graph Laplacian term of the form $\theta_0 \mathbf{x} + \theta_1 \mathbf{Lx}$. More recently, Joshi (2025) laid out transformer attention matrices as fully connected graph adjacencies to relate transformers to graph attention networks of Veličković et al. (2018).

Our interpretation of the weight matrices as learning rotation and step size suggests that transformers learn to learn or learn to optimize quickly (i.e., perform optimization in a latent variable model with just $n_{\text{blocks}}$ steps), which is a well studied field; an overview of the field of learning-to-optimize and its major ideas is presented in Chen et al. (2021).

## 2   Background

We recap ProbDR's variational Laplacian Eigenmaps formulation, which forms the basis of our interpretation. Laplacian Eigenmaps is a dimensionality reduction algorithm introduced by Belkin & Niyogi (2001) that reduces the size of a dataset $\mathbf{Y} \in \mathbb{R}^{n \times d}$ to a smaller matrix of representations $\mathbf{X} \in \mathbb{R}^{n \times d_q}, d_q << d$. The probabilistic Laplacian Eigenmaps model is a probabilistic interpretation of the algorithm (i.e. a model, inference within which leads to the algorithm in question). It can be written as follows, where a Wishart distribution is placed on a precision matrix, of which the graph Laplacian $\mathbf{L}$ is an estimate,

$$d \cdot \mathbf{L}(\mathbf{Y}) \sim \mathcal{W}((\mathbf{XX}^T + \beta\mathbf{I})^{-1}, d).$$

MAP inference for latent embeddings $\mathbf{X} \in \mathbb{R}^{n, d_q}$ in this model is equivalent to KL minimization over a random variable $\mathbf{\Gamma}$, where the model and variational constraints are written as,

$$\log p(\mathbf{\Gamma}) = \log \mathcal{W}(\mathbf{\Gamma}|(\mathbf{XX}^T + \beta\mathbf{I})^{-1}, d), \qquad \log q(\mathbf{\Gamma}) = \log \mathcal{W}(\mathbf{\Gamma}|\mathbf{L}(\mathbf{Y}), d),$$

where $\mathbf{L}(\mathbf{Y}) \in S_+^n$ is a graph Laplacian[1] matrix encoding a k-nearest neighbour graph, calculated using the data $\mathbf{Y}$. The model graphs are shown in the footnote[2]. It was shown in Ravuri et al. (2023) that the maximum of ELBO, which simplifies as $-\text{KL}(q(\mathbf{\Gamma})\|p(\mathbf{\Gamma}))$, is attained when the latent embeddings are estimated as follows,

$$\hat{\mathbf{X}} = \mathbf{U}_{d_q}\left(\mathbf{\Lambda}_{d_q}^{-1} - \beta\mathbf{I}_{d_q}\right)^{1/2}\mathbf{R},$$

where $\mathbf{U}_{d_q}$ are the $d_q$ eigenvectors of the graph Laplacian corresponding to the smallest non-zero eigenvalues encoded within the diagonal matrix $\mathbf{\Lambda}$, and where $\mathbf{R} \in O(n)$ is an arbitrary rotation matrix. Further, note that, with an additional constraint, namely $\mathbf{X}^\top\mathbf{X} = \mathbf{I}$, the optimal estimate becomes[3],

$$\hat{\mathbf{X}} = \mathbf{U}_{d_q}\mathbf{R}.$$

In the later case, assuming that the empirical mean of the embeddings is zero, the empirical variance is equal to $\sum_k \hat{\mathbf{X}}_{kj}^2/n = 1/n$.

---

[1]We denote the adjacency matrix as $\mathbf{A}$, hence $\mathbf{L} = \mathbf{D} - \mathbf{A}$, with $\mathbf{D}_{ii} = \sum_k \mathbf{A}_{ik}$.

[2]The model graph can be drawn as: (x)⟶(r) and the variational graph as: (Y)⟶(r) .

[3]This is a consequence of the trace minimisation theorem, as the objective is simply $\text{tr}(\mathbf{LXX}^T)$. Any arbitrary rotation still remains a solution as the objective and the constraint are invariant to rotations.

**The variational interpretation of SimSiam, a Semi-Supervised Learning method**

We make a short digression to show how the model graph of ProbDR is not atypical in the representation learning field. Let $\mathbf{Y}_i^a, \mathbf{Y}_i^b, ...$ be augmentations/views/modalities of a data point. SimSiam, introduced in Chen & He (2020), is a semi-supervised learning method that constructs representations of the data by minimising the negative inner product,

$$\mathcal{L}_i = - \sum_{m_a, m_b} f(h(\mathbf{Y}_i^{m_a}))^\top f(\mathbf{Y}_i^{m_b}),$$

where the element in red is under stop-grad, and with $f(\mathbf{Y}_i^m), f(h(\mathbf{Y}_i^m)) \in \mathcal{S}^{d_q-1}$. Nakamura et al. (2023) show that this loss function has a variational interpretation, where,

$$p(\mathbf{X}_i|\mathbf{Y}_i) \propto \prod_m \text{vMF}(\mathbf{X}_i|f(h(\mathbf{Y}_i^m)), \kappa), \qquad q(\mathbf{X}_i|\mathbf{Y}_i) \propto \sum_m \delta(\mathbf{X}_i|f(\mathbf{Y}_i^m))$$

$$\Rightarrow \text{KL}(q||p) = c - \mathbb{E}_{q(\mathbf{X}_i|\mathbf{Y}_i)}(\log p(\mathbf{X}_i|\mathbf{Y}_i)) = - \sum_{m_a, m_b} f(h(\mathbf{Y}_i^{m_a}))^\top f(\mathbf{Y}_i^{m_b}) = \mathcal{L}_i.$$

Due to the stop-grad applied to the elements of the loss that form the variational constraint, we note that the model graphs are very similar to ProbDR, in that the variational constraint is treated as an observed random variable. We see the variational constraint as approximating a reasonable embedding of the data *at every iteration* of the optimisation process. As an example, if $f$ were initialised as a random projection of the data, it is known that certain properties of the data are retained in the resulting embedding (due to the Johnson–Lindenstrauss lemma). If an optimisation step corresponding to the model preserves/improves these properties (and does not make $f$ degenerate or collapse), we can rely on the variational constraint to always provide an approximate but valid "view" of the data for the model to approach. We apply a similar principle in section 3.

**Transformers as unrolled optimisation**

We now summarise the idea of Yu et al. (2023) on how transformers correspond to unrolled optimisation. Assume a random variable $\mathbf{X} \in \mathbb{R}^{n \times d_q}$, where $d_q$ is the number of latent dimensions and $n$ is the number of i.i.d. data points (of image patches, text tokens, high-dimensional signals, etc.) to which rows of the representations $\mathbf{X}$ correspond. Assuming a latent variable model on $\mathbf{X}$ and a corresponding probabilistic objective $\mathcal{L}$ (a negative log density $-\log p(\mathbf{X})$ or an upper bound on it), gradient descent with $m$ steps of the objective can be unrolled as a sequence of random variables,

$$\mathbf{X}_1 \xrightarrow{T} \mathbf{X}_2 \xrightarrow{T} ... \xrightarrow{T} \mathbf{X}_s.$$

Yu et al. (2023) showed that the gradient descent operation $T$ is very similar to the operations that occur in an (encoder) transformer block, assuming a Gaussian mixture model with a sparse prior on the latent representations $\mathbf{X}$. We note that due to the representations being latent, the model considered in Yu et al. (2023) can also be thought of as a mixture of principal component analysers[4], therefore suggesting that transformers perform linear (non-kernelized) dimensionality reduction.

## 3 Transformers as ProbDR Inference

In this work, we present an alternative interpretation to that of Yu et al. (2023), that shows that transformers perform gradient descent on a variational objective derived using a variational form of the probabilistic Laplacian Eigenmaps model of Ravuri et al. (2023). We rewrite the random variable corresponding to latents as $\mathbf{Z}$, and treat $\mathbf{X}$ as a parameter that encodes latent positions. We modify the model by adding a prior on the latents,

$$\log p(\mathbf{\Gamma}, \mathbf{Z}) = \log \mathcal{W}\left(\mathbf{\Gamma}|(\mathbf{Z}\mathbf{Z}^\top + \beta\mathbf{I})^{-1}\right), d) + \log \mathcal{U}^*(\mathbf{Z}).$$

$\mathcal{U}^*$ is a matrix von-Mises-Fisher distribution (a uniform over matrices, with rows that lie on a $d_q$-dimensional hypersphere), with an additional constraint that for every row $\mathbf{x}$, $\sum_i^{d_q} x_i = 0$ (the rows have zero mean, and hence the coordinates lie on a hyperplane). Projected optimisation with this prior will lead to LayerNorm steps during optimisation.

---

[4]in a dual sense—acting on the latents and not the components.

We force the random variable $\mathbf{Z}$ to take values $\mathbf{X}$ a.s., and we modify the calculation of the graph Laplacian used in the variational constraint, so that it's a function of the latents $\mathbf{Z}$ and not the data $\mathbf{Y}$,

$$q(\mathbf{\Gamma}, \mathbf{Z}) = \mathcal{W}(\mathbf{\Gamma}|\tilde{\mathbf{L}}(\mathbf{Z}), d) * \delta(\mathbf{Z}|\mathbf{X}).$$

The graph Laplacian is computed as $\tilde{\mathbf{L}} = \mathbf{I} - \tilde{\mathbf{A}}(\mathbf{Z}) = \mathbf{I} - \sigma(\kappa \mathbf{Z}\mathbf{Z}^T - \mathbf{M})$ where $\sigma$ is the softmax function, applied row-wise (so that the row sums of the input matrix all equal one). $\tilde{\mathbf{A}}$, we argue, is a soft (differentiable) proxy to the true nearest neighbour adjacency matrix, particularly when the latent embeddings $\mathbf{X}$ are initialised with PCA or random projections, as $\mathbf{X}\mathbf{X}^\top$ is a minimal-error estimate of the empirical covariance of the data, and the covariance between similar points is expected to be similar in value. This leads to the row-wise softmax being similar and high for similar points, encoding a similarity structure. $\mathbf{M}$ is a mask matrix (for example, if we were to disallow self-adjacency, $\mathbf{M}$ can be set to $\iota\mathbf{I}$, with $\iota \to \infty$), and $\kappa$ is a hyperparameter that can be tuned such that the proxy adjacency $\tilde{\mathbf{A}}$ is "close to" a reference nearest neighbour matrix.

In a similar fashion to ProbDR, and the variational interpretation to SimSiam, we treat the variational constraint as an observed random variable, and hence do not account for gradient updates to terms leading from the variational constraint. Hence, the KL-div. with stop-grad applied to the variational constraint is,

$$\mathrm{KL}\big(q(\mathbf{\Gamma}, \mathbf{Z})\|p(\mathbf{\Gamma}, \mathbf{Z})\big) \propto \underbrace{\mathrm{tr}\big(\tilde{\mathbf{L}}(\mathbf{X}\mathbf{X}^\top + \beta\mathbf{I})\big)}_{\mathcal{L}_{\mathrm{data}}} - \underbrace{\log\det(\mathbf{X}\mathbf{X}^\top + \beta\mathbf{I})}_{\mathcal{L}_{\mathrm{reg}}} + c,$$

where $\forall i : \mathbf{X}_i \in \mathcal{S}^{d_q - 1}$ and $\sum_j \mathbf{X}_{ij} = 0$. Yu et al. (2023) show that a transformer block's sequence of updates follows gradient descent of an objective in steps; given an objective $\mathcal{L}(\mathbf{X}) = \mathcal{L}_{\mathrm{data}}(\mathbf{X}) + \mathcal{L}_{\mathrm{reg}}(\mathbf{X})$, they interpret a transformer block calculations as an alternating optimisation process involving the updates,

$$\mathbf{X}' \longleftarrow \mathbf{X} - \eta * \frac{d\mathcal{L}_{\mathrm{data}}}{d\mathbf{X}}, \qquad \mathbf{X} \longleftarrow \mathbf{X}' - \eta * \frac{d\mathcal{L}_{\mathrm{reg}}}{d\mathbf{X}'}.$$

In this work, we analyze the transformer at initialization (e.g., with all weights set to diagonal matrices) and consider transformers with single heads, which simplifies the analysis for exposition. We believe that this can be trivially extended by considering a product-of-experts type distribution as part of the variational constraint. Furthermore, for this work, we ignore the ReLU activation that is part of the fully connected segment of the transformer for ease of exposition; however, this can be re-added simply by incorporating a sparsity prior, derived in Yu et al. (2023), as our regularization term is identical to theirs (the sparsity terms notwithstanding).

We now show how an (encoder) transformer block's operations arise as optimisation steps of our objective. First, note that, $d\mathcal{L}/d\mathbf{X} = 2\tilde{\mathbf{L}}\mathbf{X} = 2(\tilde{\mathbf{A}} - \mathbf{I})$, and a gradient descent update for optimisation of $\mathcal{L}_{\mathrm{data}}$ follows,

$$\mathbf{X} \longleftarrow \mathbf{X} + 2\eta(\sigma(\kappa\mathbf{X}\mathbf{X}^\top - \beta\mathbf{I} - \mathbf{M}) - \textcolor{red}{\mathbf{I}})\mathbf{X}.$$

The element highlighted (which is the degree matrix, in this case, the identity matrix) in red shows the only difference to a standard attention operation (as the attention matrix is the only term that appears in the ordinary architecture). Next, we must take a projection step to ensure that $\forall i : \mathbf{X}_i \in \mathcal{S}^{d_q - 1}$ and $\sum_j \mathbf{X}_{ij} = 0$, and hence,

$$\mathbf{X} \longleftarrow \mathrm{LayerNorm}(\mathbf{X}).$$

We now optimise w.r.t. $\mathcal{L}_{\mathrm{reg}}$. Note that this is exactly the same form of regularisation (apart from the sparse prior that gives rise to the ReLU, which is ignored for the sake of exposition, but can trivially be introduced) as the term that appears in Yu et al. (2023). We refer the reader to that work for a careful argument for how this term approximately gives rise to a linear update, but here, we simply approximate $d\mathcal{L}_{\mathrm{reg}}/d\mathbf{X} = 2(\mathbf{X}\mathbf{X}^T + \beta\mathbf{I})^{-1}\mathbf{X} \approx 2/(d_q + \beta)\mathbf{X}$, and our remaining optimisation steps simply involve this update and another projection,

$$\mathbf{X} \longleftarrow \mathbf{X} - \frac{2\eta}{\beta + d_q}\mathbf{X}$$

$$\mathbf{X} \longleftarrow \mathrm{LayerNorm}(\mathbf{X}),$$

which completes the transformer block operations, assuming simple initialisations. Note that a key insight is that the probabilistic interpretation differs from practice in that the former does Laplacian smoothing (**graph diffusion** - i.e. the subtraction of an identity matrix, or a degree matrix, from the attention matrix), whereas the later does not.

**An interpretation of the weight matrices**

We posit that an update such as $\mathbf{X} \longleftarrow \mathbf{X} + \mathbf{X}\mathbf{W}_{\text{lin}}$ can be interpret as a rotation (which, under the probabilistic Laplacian Eigenmaps model, the solution is invariant to) and a scaling, which, under our interpretation, corresponds to a learnt step size $\eta = |\mathbf{W}|^{1/d_q}$; this is a restatement of the belief that transformers *learn to learn*, in other words, perform optimisation (assuming a dimensionality reduction or clustering model) with few steps.

## 4    Experiments

We provide three main experiments to show validity of the ideas presented thus far. In the first, we show that a transformer initialised in a simple way performs dimensionality reduction, using flattened images from the MNIST dataset. In the second, we show that removing an identity matrix from the attention matrix as suggested by our derivations increases performance on the Shakespeare dataset and a downsampled (16-by-16) version of Imagenet. It can also be shown that training of GPT-2 converges faster with our modification, than without.

### 4.1    Transformers perform Dimensionality Reduction

The details of our dimensionality reduction experiment are as follows. We set up a sequential neural network, with an initial projection layer with weight $\mathbf{W}_{\text{proj}} \sim \mathcal{MN}(0, \mathbf{I}_d/d, \mathbf{I}_{d_q})$ that is randomly initialised with Gaussian entries (i.e. a Gaussian random projection). Next, the network consists of a set of $n_{\text{blocks}} = 8$ encoder transformer blocks. We found that increasing the number of blocks makes the latents collapse into extremely tight clusters. The LayerNorms have post-normalization weights associated with them, $\mathbf{W}_{\text{norm}} = \mathbf{I}/\sqrt{n}$, which is because we expect the optimum to be akin to eigenvectors of a graph Laplacian, which would have variance $1/n$, as explained in the background. The transformer block weights are $\mathbf{W}_{\text{q}} = \sqrt{\kappa n}\mathbf{I}, \mathbf{W}_{\text{k}} = \sqrt{\kappa n/q}\mathbf{I}$ and $\mathbf{W}_{\text{v}} = 2\eta$ corresponding to the query, key, value weight matrices. The query and key matrices were set up such that the attention matrix, pre-normalisation, has a diagonal equal to $\kappa$. We set $\kappa = 30$, based on the clustering empirically observed in the resulting graph Laplacian's eigenvectors. Finally, the feed-forward block is a single layer with weight $\mathbf{W}_{\text{lin}} = -2\eta$. Note that, based on our derivation (specifically, the scalar coefficient to the attention matrix), $\eta$ can be a maximum of $0.5$ to avoid magnitudes of the updates being too large, and so we use the learning rate $\eta = 0.4$. We use the latent dimension $d_q = 128$. Passing the flattened images through the transformer can be seen to perform clustering, as illustrated in fig. 1.

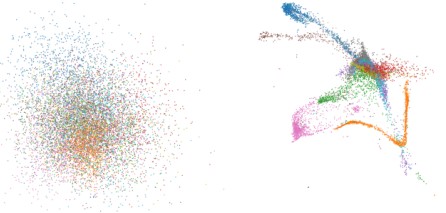

Figure 1: The first two latent dimensions corresponding to flattened MNIST images after a random initialisation (i.e. the initial random projection layer that converts pixels to a latent representation) (**left**), and after eight steps through a transformer block (**right**), showing that transformer blocks cluster points in the latent space.

### 4.2    Graph Diffusion improves performance

In the second experiment, we simply replace the attention matrix $\mathbf{A}$ within a transformer architecture, found in nanoGPT (Karpathy, 2022) with the negative graph Laplacian $\mathbf{A} - \mathbf{I}$, and run the model multiple times on the Shakespeare dataset. We also repurpose the code to build a small vision transformer, and train it naively (i.e. without random augmentations, learning rate schedules, etc.) on the downsampled Imagenet dataset, wherein all images are 16 by 16 pixels. On this dataset, a

benchmark given in Chrabaszcz et al. (2017) achieves 40% validation accuracy, whereas our naïve ViT acheives around 26%. In both cases however, validation performance improves when we replace the attention matrix by the negative graph Laplacian.

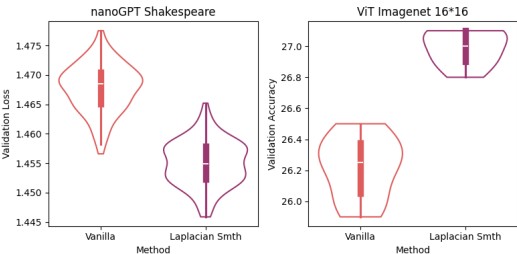

Figure 2: **Left:** validation losses on the Shakespeare dataset and **right:** validation accuracies on a downsampled Imagenet dataset, showing that Laplacian smoothing achieves a better performance in both cases.

## 5  Conclusion

We have shown that transformer blocks correspond to unrolled inference assuming a probabilistic Laplacian Eigenmaps model, and that a simple architectural tweak—using a negative Laplacian $\mathbf{A} - \mathbf{I}$ in place of the attention matrix $\mathbf{A}$—yields consistent gains in language and vision settings. Future work will make more careful approximations of the ideas presented, expand on the experimental validation (current limitations of the work), explore whether non-linear (kernelized) probabilistic models of dimensionality reduction (from Ravuri & Lawrence (2024)[5]) can increase performance in models with lower latent dimensionality, and relate transformers to other generalized architectures. Code used for this paper can be found at this link (note that, for the GPT experiments, this is a very simple modification of Karpathy (2022)).

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
