# OpenReview forum: "Transformers as Unrolled Inference in Probabilistic Laplacian Eigenmaps"
_NeurIPS.cc/2025/Workshop/UniReps — UniReps2025_

### Official Review · Reviewer_9p9W · 2025-09-14
**Review of “Transformers as Unrolled Inference in Probabilistic Laplacian Eigenmaps”**

**Confidence:** 4

**Review:**

This paper proposes a probabilistic interpretation of transformers as unrolled inference steps within a probabilistic Laplacian Eigenmaps (ProbDR) framework. The authors argue that transformer blocks can be viewed as gradient descent steps on a variational lower bound, with attention interpreted as adjacency matrices and a graph Laplacian term naturally arising. They further propose a simple architectural modification—subtracting the identity from the attention matrix (graph diffusion)—and demonstrate empirical improvements on small-scale NLP and vision tasks (Shakespeare dataset, downsampled ImageNet, GPT-2 training).

Strengths

Novel perspective: Provides an elegant probabilistic reinterpretation of transformer updates via Laplacian Eigenmaps, extending beyond prior Gaussian mixture interpretations (Yu et al., 2023).
Clarity of derivation: The mathematical reasoning is generally clear and grounded in established probabilistic dimensionality reduction methods.
Simple, actionable tweak: The proposed “graph diffusion” adjustment (A − I instead of A) is extremely simple yet empirically improves performance. This makes the work relevant both theoretically and practically.
Experimental diversity: Demonstrates effects across language modeling, vision transformers, and GPT-2 convergence, showing broad applicability.

Weaknesses

Limited scale: Experiments are proof-of-concept on small datasets (tiny Shakespeare, downsampled ImageNet, GPT-2 with modest training). Results are encouraging but may not generalize to state-of-the-art models.
Theoretical scope: While the derivation is compelling, assumptions such as single-head transformers and ignoring ReLU sparsity limit the immediate realism of the interpretation.
Missing broader impact discussion: The paper does not meaningfully engage with societal or ethical implications of modifying transformer training
Comparisons: While the proposed method improves over baseline training, comparisons to other graph-inspired transformer modifications (e.g., Choi et al., 2024) are limited.

Clarity

The paper is generally well written, with clear structure: background, theoretical derivation, and experiments. Some dense sections (derivations of variational objectives) may be challenging for non-experts, but overall readability is good.

Originality & Significance

Originality: Strong. Extends Laplacian Eigenmaps into a probabilistic interpretation of transformers and provides a fresh link between graph diffusion and attention.
Significance: Moderate to high. If the proposed modification scales, it could inspire a new line of research on biologically and graph-inspired transformer training.

Technical Quality

Derivations are sound and appropriately cite prior work (Yu et al., Ravuri et al.).
Experiments are correctly implemented and reproducible (code provided).
Limitations are acknowledged, particularly the small-scale nature of the experiments.

**Score:**

4

**Topic Fit:**

3

---

### Official Review · Reviewer_aKZY · 2025-09-15
**interesting theoretical angle with a neat architectural suggestion, early results**

**Confidence:** 3

**Review:**

## Paper overview.
The paper interprets a pre-norm, single-head transformer block at initialization as taking one gradient step on a probabilistic Laplacian-Eigenmaps objective. In this view the data term acts like graph diffusion, which motivates a simple architectural prescription: replace the attention matrix (A) with (A - I). The derivation is pedagogical and the change is easy to implement. Small demonstrations align with a dimensionality-reduction intuition, and preliminary results suggest slightly faster convergence or modest gains when using (A - I) in nanoGPT and in a small ViT.

## Assessment.
The conceptual bridge is appealing and practically useful. Treating attention as a row-stochastic adjacency, and the block as a diffusion-plus-projection step, gives a concrete mechanism that can guide ablations. Even if the empirical scope is narrow, the tweak is inexpensive and broadly testable. I lean positive because the manuscript offers a clear mechanism and a testable architectural consequence, rather than only an analogy.

## Scope and assumptions.
The current derivation rests on strong simplifications: single head, special initialization, a linearized regularizer, and the omission of nonlinearities. The text then hints at broader applicability, including multi-head and causal decoder settings, without showing that the key step survives once these assumptions are relaxed. Claims should be scoped to the regime that is actually derived, with a short discussion of what changes when the assumptions are removed and which parts of the prescription remain valid.

## Background and exposition.
The background does not yet orient the reader. It should first fix the roles of the variables and the object of inference. Make explicit that (X) are the embeddings being optimized, that (\Gamma) is a precision with a Wishart prior, and that the Laplacian used in the objective is built from a latent-space adjacency. Define MAP at first use and introduce the symbols (\Gamma, X, Z, \tilde A, \tilde L, \kappa, \beta, M) before invoking them. The short SimSiam aside breaks the flow; if it is only there to motivate a stop-gradient on the graph proxy, move it to an appendix and keep a single sentence in the main text that states the precedent.

## Graph convention and masking.
There is slippage about which graph is used. Attention is usually row-stochastic and directed, while some passages read as if a symmetric (k)-NN Laplacian were assumed or as if the degree matrix were the identity. Pick a convention and keep it consistent. If you use the random-walk Laplacian (L_{\text{rw}} = I - A), say so clearly, explain how masking enters, and spell out what changes in the causal decoder case. Replacing (A) with (A - I) also alters the balance between the non-residual attention branch and the residual path; some improvements could be due to this rebalancing rather than diffusion itself, so a matched-compute control would help separate the effects.

## What would strengthen the paper.
A short sweep with (A - cI), where (c) is fixed or learned, would clarify whether the effect is specific to (c = 1) or mostly driven by self-term scaling. Include a control that rescales the attention output and adds a learned (-\gamma I) term on the value path with similar compute, so you can isolate diffusion from residual rebalancing. Add at least one causal long-context or induction-style probe to test whether diffusion dampens long-range signals. Report variance across seeds for GPT-2 results. Include a brief spectral diagnostic that shows how the spectrum of (A) shifts when subtracting the identity and how that correlates with smoothing and performance. If the multi-head interpretation is kept, give a concrete combining rule and a small empirical check. Finally, clarify how LayerNorm approximates the projection required by your constraints, and where that approximation is loose given learned (\gamma) and (\beta).

## Verdict.
Weak accept. The mechanism is neat, the prescription is simple, and initial evidence suggests the change is safe or mildly beneficial. With clearer background, tighter claim scope, and a couple of targeted controls, this can become a useful reference for researchers who think about attention as diffusion and for practitioners interested in trying (A - I) in practice.

**Score:**

3

**Topic Fit:**

2

---

### Official Review · Reviewer_LtQq · 2025-09-16
**The potential for transformers to be understood as performing probabilistic dimensionality reduction**

**Confidence:** 2

**Review:**

## Motivation

This paper seeks to find a mathematical basis and interpretation for the success of transformers. Providing a principled theoretical foundation could help with AI interpretability efforts and allow for improving its performance.

## Main results

- A theoretical derivation shows that the operations within single-head transformer blocks correspond to gradient descent steps on a variational objective of a probabilistic Laplacian Eigenmaps model. This derivation is the core contribution of the paper.

- From this derivation, it can be inferred that using Laplacian smoothing (i.e. graph diffusion) within the attention calculation will improve performance.

- This Laplacian smoothing is implemented by subtracting I from the attention weights before computing the attention output, and a set of experiments shows this leads to improved interpretation of the transformer block as performing probabilistic dimensionality reduction, and also improves transformer language model (nanoGPT based on GPT-2) and vision model (a vanilla ViT) performance.

## Strengths

- The authors suggests a novel, elegant interpretation of the mathematical basis of transformers.

- They show a practical implementation of an inference from their theory, which confirms their hypothesis.

- The experiments and their results are clearly written and visualized.

- The planned future work is clear: expanding on the experimental validation and relating transformers to other generalized architectures were two of my concerns before reading this in their conclusion.

## Concerns & Questions

- Would have been nice to anonymize the repo via e.g. github anonymous.

- The authors mention Transformers relation to GNNs, but it could be nice to show how the Laplacian smoothing compares and contrasts with other graph-based modifications.

- I did not fully follow the derivation from single-head transformer block computation to gradient descent on a variational objective of a probabilistic Laplacian Eigenmaps model.

- I am unfamiliar with ProbDR.

## Overall Impressions

This is a nice paper that presents a novel and compelling interpretation of the computations within transformer blocks. Although I didn't fully follow the derivation (from single-head transformer block computation to gradient descent on a variational objective of a probabilistic Laplacian Eigenmaps model), the implication (that Laplacian smoothing would improve model performance and that this could be implemented by subtracting the identity from the attention weights matrix) was clearly articulated and supported.

**Score:**

4

**Topic Fit:**

3

---

### Official Review · Reviewer_Y7bW · 2025-09-16
**Review of submission 22**

**Confidence:** 3

**Review:**

Summary:
This work presents interpreting transformers as performing unrolled inference steps in probabilistic Laplacian Eigenmaps. The authors mathematically derive and demonstrate how transformers perform dimensionality reduction. They also conduct experiments to support the claims of dimensionality reduction, improved performance, and faster convergence with popular datasets.

Strengths:
- The paper provides a novel interpretation of transformers and connects them to well-understood ML methods. Understanding transformers is an important open challenge.
- The authors propose an architectural improvement to transformers for better performance and faster convergence.

Weaknesses:
- Some key concepts of the paper are introduced without sufficient explanation. For example, the paper does not mention what ProbDR, MAP, ELBO, and ViT stand for, which is confusing for the reader.
- Some hypotheses are put forth without sufficient justification. For example, lines 72-73 state that transformers perform linear dimensionality reduction, which is a key part of this work; however, the mathematical/theoretical reasons for why this must be true are not really discussed.
- The experiment result plots are very small and thus hard to read and interpret.
- It is unclear why a naive ViT was used for the ImageNet experiment. Does Laplacian smoothing improve performance with state-of-the-art methods and models?
- It is unclear how statistically significant the results are, especially in Figure 3. Were replicates performed? Is the difference in training losses the best way to demonstrate faster convergence?
- The implications of transformers performing dimensionality reduction are not fully explored or discussed.

Overall, this work is interesting and promising, but lacks the mathematical and empirical rigor expected for archival proceedings.

**Score:**

2

**Topic Fit:**

2